# Identification and Evaluation of Novel Antigen Candidates against *Salmonella Pullorum* Infection Using Reverse Vaccinology

**DOI:** 10.3390/vaccines11040865

**Published:** 2023-04-18

**Authors:** Zhijie Jiang, Xiamei Kang, Yan Song, Xiao Zhou, Min Yue

**Affiliations:** Institute of Preventive Veterinary Sciences, Department of Veterinary Medicine, College of Animal Sciences, Zhejiang University, Hangzhou 310058, China; jiangzhijie@zju.edu.cn (Z.J.);

**Keywords:** *S. Pullorum*, PstS, reverse vaccinology, chick infection model, immunogenicity

## Abstract

Pullorum disease, caused by the *Salmonella enterica* serovar Gallinarum biovar Pullorum, is a highly contagious disease in the poultry industry, leading to significant economic losses in many developing countries. Due to the emergence of multidrug-resistant (MDR) strains, immediate attention is required to prevent their endemics and global spreading. To mitigate the prevalence of MDR *Salmonella Pullorum* infections in poultry farms, it is urgent to develop effective vaccines. Reverse vaccinology (RV) is a promising approach using expressed genomic sequences to find new vaccine targets. The present study used the RV approach to identify new antigen candidates against Pullorum disease. Initial epidemiological investigation and virulent assays were conducted to select strain R51 for presentative and general importance. An additional complete genome sequence (4.7 Mb) for R51 was resolved using the Pacbio RS II platform. The proteome of *Salmonella Pullorum* was analyzed to predict outer membrane and extracellular proteins, and was further selected for evaluating transmembrane domains, protein prevalence, antigenicity, and solubility. Twenty-two high-scored proteins were identified among 4713 proteins, with 18 recombinant proteins successfully expressed and purified. The chick embryo model was used to assess protection efficacy, in which vaccine candidates were injected into 18-day-old chick embryos for in vivo immunogenicity and protective effects. The results showed that the PstS, SinH, LpfB, and SthB vaccine candidates were able to elicit a significant immune response. Particularly, PstS confers a significant protective effect, with a 75% survival rate compared to 31.25% for the PBS control group, confirming that identified antigens can be promising targets against *Salmonella Pullorum* infection. Thus, we offer RV to discover novel effective antigens in an important veterinary infectious agent with high priority.

## 1. Introduction

The *Salmonella enterica* serovar Gallinarum biovar Pullorum (*Salmonella Pullorum*), a Gram-negative poultry-restricted pathogenic bacterium, causes well-recognized Pullorum disease (PD) [1,2]. Pullorum disease is an acute systemic disease that mainly affects chicks under three weeks old [3], leading to septicemia and white viscous diarrhea with high morbidity and mortality [4]. In adult birds, *S. Pullorum* infection can result in weight loss, decreased fertility and laying, diarrhea, and reproductive tract abnormalities with a chronic carrier state [5]. *S. Pullorum* is challenging to control due to its vertical and horizontal transmission capabilities [6,7,8]. It has become more prevalent recently and poses a severe threat in many developing countries [9]. Antibiotic treatment can reduce the risk of disease, but the emergence of multidrug-resistant (MDR) strains of *Salmonella* has raised concerns about the effectiveness of antibiotic treatments, and it represents a significant threat to the poultry industry [10,11,12,13,14,15,16,17,18,19]. Therefore, developing and implementing effective vaccines to control and prevent PD is a priority for many developing countries.

Current *Salmonella* vaccine candidates have concerns regarding insufficient attenuation or low immunogenicity [20]. There are two licensed vaccines, including a live attenuated *S. Gallinarum* 9R strain (Nobilis^®^ SG 9R) [21] and Salenvac^®^ (Merck, Rahway, NJ, USA), a killed *S. Enteritidis* vaccine strain grown under iron-limiting conditions [22]. Live vaccines, although effective in providing immunity, can also regain their virulence, thereby increasing the risk of further environmental contamination and endangering the flock’s health [23,24]. On the other hand, killed vaccines are generally considered safe, but they often fail to induce a sufficient protective effect against the disease. Subunit vaccines consist of a single antigen or multiple defined antigens, and are well-recognized as an excellent candidate. These antigenic components are commonly present on the surfaces of bacterial cells and play a role in virulence [25,26,27,28]. Research on *Salmonella* subunit vaccine candidates, including SseB [29], FliC [30], OmpD [31], OmpC [32], and PagN [33], was conducted. However, none of these could provide appropriate immunogenicity or broad protection against *Salmonella* infections. Therefore, continuous efforts are being undertaken to discover new vaccine antigens.

The development of integrated bioinformatics and immunological informatics approaches, as well as the accumulation of bacterial genomic sequences and protein structures, have sped up and reduced the cost of vaccine development in recent years [34]. Reverse vaccinology (RV), a novel and potent in silico prediction approach for identifying new protein-based vaccine candidates [35], overcomes the current vaccinology limitation [36]. The first successful experience of the reverse vaccinology approach was used to detect vaccine candidates against *Neisseria meningitidis* serogroup B (MenB) [37] and eventually developed a general MenB vaccine [38]. Computational in silico prediction tools were used to examine all proteins’ physical–chemical characteristics to screen potential immunogens. For example, the subcellular locations of proteins were predicted to select surface-exposed proteins as the candidates [25]. In addition, proteins with adhesive capabilities are known to be involved in bacterial pathogenicity and invasion, so adhesins or adhesin-like proteins can be used as potential vaccine targets [39]. The number of transmembrane helices is a limiting factor, as proteins with multiple transmembrane helices are usually challenging to purify [37]. Here, the high-throughput screening of *S. Pullorum* surface proteins was performed to identify novel conserved antigens, using an extensive collection of clinical isolates from a nationwide epidemiological investigation. Additional immunogenicity and the protective efficacy of the optimal antigen candidates were further investigated in chick models.

## 2. Materials and Methods

### 2.1. S. Pullorum Strain R51 Isolation and Identification

The strain *S. Pullorum* R51 was isolated from the liver of a sick chicken on a farm located in Anhui, China, which was densely covered with white liver spots. The isolation method was performed as described before [40]. Finally, the single colony grown on a xylose lysine desoxycholate agar (XLD, Land Bridge, Beijing, China) plate was identified via polymerase chain reaction (PCR) using two pairs of primers, *invA*-F/R (*S. Pullorum*/*S. Gallinarum* positive: 517 bp) and *ratA*-F/R (*S. Pullorum* positive: 243 bp; Negative: 1047 bp) (Appendix A).

### 2.2. Virulence Assays in Selecting the Candidate Strain

Animal experiments were conducted under the approval and supervision of the Zhejiang University Animal Ethics Committee. To assess the virulence of strain R51, seven *S. Pullorum* isolates (Appendix A) were randomly selected to conduct the chick embryo infection and chick infection experiments. The overnight cultured bacterial liquid suspension was diluted with LB broth to 10^9^ CFU/mL (OD_600nm_: 0.5), and then 10-fold diluted with PBS to 10^4^ CFU/mL (plating on LB solid agar, followed by colony counting). Each 16-day-old chick embryo in the infection and control group was inoculated with 100 μL bacteria and PBS via allantoic cavity injection, respectively, and monitored daily until hatching.

For the chick infection, the chick feces were collected to confirm the absence of *Salmonella*. The concentration of overnight bacteria was adjusted to 5 × 10^9^ CFU/mL. One-day-old chicks were orally infected with 200 μL bacteria with 30 chicks in each group, and observed for 14 consecutive days to monitor their vitality.

### 2.3. Whole Genomic Sequencing and Additional Clinical Collections

The genomic DNA of R51 was extracted using the SDS-based DNA extraction method [41], whose quality and quantity were evaluated using agarose gel electrophoresis and a Qubit 2.0 fluorometer. A single-molecule real-time (SMRT) sequencing library with an insert size of 10 kb was generated using the SMRT bell Template kit v.1.0 (Pacific Biosciences of California, Shanghai, China), and a high-quality Illumina sequencing library was prepared using the NEBNext Ultra DNA Library Prep Kit (New England Biolabs, Beijing, China). The whole-genome sequencing of R51 was accomplished using the PacBio Sequel platform and Illumina NovaSeq PE150 by Novogene Technology Co., Ltd., Beijing, China, data upload NCBI GeneBank, login number: nz_cp068386.1.

The proteomic sequences of *S. Pullorum* R51 have been subjected to different servers to identify the potential antigenic targets. The whole proteomes of 182 other *S. Pullorum* strains were retrieved in our laboratory (Appendix A). The in silico analytic approaches used in the present study are summarized in Table 1.

### 2.4. Selection of Essential Proteins

Pathogens rely on essential proteins to survive within a host. Among these proteins, surface-exposed ones, such as outer membrane and extracellular proteins, are easily accessible to the immune system’s antigen-presenting cells. Consequently, they make excellent vaccine candidates. To predict the subcellular localization of all proteins in the *S. Pullorum* R51’s proteome, four dedicated online servers were utilized: PSORTb v3.02 [42], BUSCA [43], SOSUI-GramN [44], and CELLO2GO [45]. Proteins considered as being surface-exposed by at least one of these servers were selected for further analysis.

### 2.5. Identification of the Signal Peptide

The signal peptide is typically a 20–40 amino acid extension at the amino terminal. It provides information about proteins that were initially designated as membranes, and distinguished between secreted and cytosolic proteins. Signal peptidases remove the signal peptide during or after translocation [46]. SignalP5.0 was used to predict cleavage sites and to indicate signal/non-signal peptides in proteins [47,48]. Proteins that contain a signal peptide will serve as the foundation for the further analysis of vaccine targets.

### 2.6. Antigenicity Prediction

To determine the antigenic properties of surface-exposed proteins, the VaxiJen online server was employed [49]. Only proteins with a probability of ≥0.8 were classified as protective antigens and selected for further analysis.

### 2.7. Evaluation of Adhesin Probability

Adhesins or adhesin-like proteins have been shown to play a significant role in bacterial pathogenicity and invasion, making them promising vaccine targets. To identify suitable protein candidates, the SPAAN server was utilized to analyze their adhesion probability [50]. Proteins with a prediction threshold of adhesion ≥ 0.5 were considered for further analysis.

### 2.8. Prediction of Transmembrane Domains and Solubility

To further refine the selected proteins, TMHMM v.2.0 servers [51] and SoDoPE (TISIGNER) https://tisigner.com/ (accessed 15 February 2022) [52] were used to predict transmembrane domains and solubility, respectively. Proteins with a probability of solubility ≥ 0.5 were considered as soluble and selected for further analysis, while those anticipated to be insoluble and having two or more predicted transmembrane domains were eliminated.

### 2.9. Protein Sequence Conservation Analysis

To screen out conserved proteins in other *S. Pullorum* strains, a BLASTp analysis was conducted to search for similar sequences in 182 *S. Pullorum* clinical isolates collected in a Chinese nationwide investigation (Appendix A). Among the candidate proteins, those exhibiting > 98% conservation between *S. Pullorum* strains were selected.

### 2.10. Bacterial Strains and Growth Conditions

*S. Pullorum* R51 was isolated from the liver of an infected bird. *E. coli* TG1 and Rosetta (DE3) were grown in a Luria-Bertani (LB) medium at 37 °C. Bacteria containing recombinant plasmids were cultured by adding kanamycin (50 μg/mL) to the growth medium.

### 2.11. Gene Cloning, Protein Expression, and Purification

The selected high-score proteins were subcloned into pET-30a after in silico analyses. All primers used in this study are listed in (Appendix A). Gene transformation was carried out following the protocol of the One Step Cloning Kit (Vazyme Biotech, Nanjing, China). The recombinant constructs were transformed into *E. coli* TG1 and cultured on LB agar plates containing 50 μg/mL kanamycin, and positive colonies were screened with colony PCR and further confirmed via sequencing.

Proteins were expressed in *E. coli* Rosetta via auto-induction in a high-density shaking cultures approach [53]. First, 1 mL overnight bacterial culture was added into 100 mL fresh LB liquid medium supplemented with 50 μg/mL of kanamycin. The culture was incubated at 37 °C, 180 rpm for 4–6 h until OD_600nm_: 0.6–0.8. Next, IPTG was added to a final concentration of 1 mM and protein expression was induced by incubating the culture at 16 °C, 120 rpm for 16 h. After inducing expression, the cells were harvested and sonicated on ice (150 W, sonicate 4 s, pause 4 s, 45 min). The cell lysate was centrifuged, and the pellet was solubilized in 8 M urea, followed by centrifugation. Fractions were analyzed on SDS-PAGE and Western blot by using a His-tag mouse monoclonal antibody (Beyotime, Shanghai, China) for protein expression.

The inclusion bodies dissolved in 8 M urea require gradual reduction of the urea concentration for refolding. The dialysis bag containing the inclusion bodies is successively placed in solutions of 6 M, 4 M, 2 M, 1 M, 0.5 M, and 0.1 M urea for dialysis, with each concentration being dialyzed for 8–12 h.

Protein purification was carried out using HisSep Ni-NTA agarose resin (YEASEN, Shanghai, China). After the protein was fully bound to Ni-NTA agarose in column, the column was washed with 10 mL of 50 mM PBS containing 20 mM imidazole 5 times, the target protein was eluted by adding 10 mL of 50 mM PBS containing 500 mM imidazole, and the eluted liquid was collected. The purified proteins were confirmed via SDS-PAGE analysis. The concentration was determined using the BCA method, and the proteins were finally stored at −20 °C.

### 2.12. In Ovo Vaccination with Candidate Proteins

Fertilized eggs from SPF (Ross 308) broilers were kept at 38 °C and 65–75% relative humidity in a forced-air egg incubator. On embryonic day 18, the eggs were candled to check for fertilization, and then the eggshell was disinfected by spraying 1.5% hydrogen peroxide and punctured using a sterile needle. A total of 20 μg of proteins diluted in 100 μL PBS mixed with 100 μL *Rhizoma Atractylodis Macrocephalae* polysaccharides adjuvant (1 mg/mL) [54] was injected into the amniotic cavity. The blood sample was taken from the jugular vein on day 11 post-hatching for antibody analysis.

### 2.13. Immunogenicity Assay

The indirect Enzyme-Linked Immunosorbent Assay (ELISA) was used to examine the antibody response to the injected proteins. In duplicate repeats, 2 μg of purified protein in 100 μL coating buffer was added to 96-well microtiter plates (Corning 3590) and incubated at 4 °C for 16 h. The wells were washed with 200 μL PBST (PBS containing 0.05% Tween20) three times, and then blocked with 150 μL 1% (*w*/*v*) casein for 1 h. After washing, serial serum dilutions in PBST (1/1000) were added and incubated for 2 h at 37 °C. Wells were washed, and 100 μL of diluted (1/15,000) HRP-conjugated anti-chicken IgY (Solarbio, Beijing, China) was added to wells and incubated for 2 h, followed by washing. TMB (100 μL) was added as a substrate, and the reaction was stopped with 2 M H_2_SO_4_ after color development. The optical density at 450 nm was quantified. Two negative controls were used, one with only coating buffer (Ag) and the other with only PBST and no serum (Ab).

### 2.14. Bacterial Loads after Challenge

All birds were orally inoculated with live R51 (1 × 10^7^ CFU/bird) on day 11 post-hatching. To prepare the inoculum, 100 μL overnight R51 culture was added to 100 mL LB broth and incubated until the mid-log phase was reached. The bacterial cells were then harvested via centrifugation at 8000× *g* for 20 min and washed with PBS to remove impurities. The pellet was suspended in PBS to achieve a concentration of 10^8^ CFU/mL. On day 7 post-challenge, liver and spleen samples were collected from the birds and subjected to R51 quantification using the plate counting method.

### 2.15. Immune Protection Assessment

To assess the effectiveness of the vaccine candidates, an experimental survival assay was conducted. The immunization procedure was identical to that described above. On day 11 after hatching, all birds were exposed to R51 strains at a concentration of 1 × 10^9^ CFU/bird. Chick mortality was monitored daily for a period of 14 days.

### 2.16. Statistical Analyses

In this study, GraphPad Prism 9 software was used for statistical analysis. The antibody responses of immunized and non-immunized chicks were compared using a Student’s *t*-test. The experiments were conducted in duplicate, and a significance threshold of *p* < 0.05 was set to determine statistical significance.

## 3. Results

### 3.1. Virulence Assay for the Selection of the Candidate Strain

After selecting representative *S. Pullorum* strains from our laboratory, chick embryo infection and chick infection experiments were conducted. The results indicated that the R51 and SAL00737 strains exhibited the highest toxicities in chick embryos, and R51 exhibited the highest lethality in chicks (Figure 1).

### 3.2. Complete Genome and Proteome of R51

Pacbio RSII was used to determine its complete genome sequence (Figure 2A). The in silico approach was used to identify new and potential vaccine candidates in the *S. Pullorum* R51 strain. The proteome of the R51 strain has 4713 open reading frames, and all of these proteins were used for further analysis (Appendix A). The overview of the screening process obtained is exhibited (Figure 2B).

### 3.3. OMP and Extracellular Protein Preselection

The predicted surface-exposed proteins are summarized in Table 2. Four subcellular localization prediction programs produced varying percentages of prediction. Among the 4713 proteins of *S. Pullorum* R51, SOSUI, BUSCA, CELLO2GO, and PSORTb predicted 702, 517, 910, and 344 proteins as surface-exposed, respectively (Appendix A). From these predictions, 1141 proteins were selected for further analysis, as they were predicted by at least one of the four software programs.

### 3.4. Identification of the Signal Peptide

To predict the presence of the signal peptide in proteins, the SignalP 4.1 software was used. Out of the 4713 proteins in *S. Pullorum* R51, 644 proteins were found to have a signal peptide on their N-terminus (Appendix A). The signal peptide plays a crucial role in transporting proteins to the outer membrane or periplasmic surface. Using this subtractive proteomics approach, the list was narrowed down to 579 proteins for further investigation.

### 3.5. Antigenicity Prediction

High antigenicity proteins can stimulate significant immune responses in the host. The VaxiJen v2.0 server was used to assess the antigenicity of the proteins. Out of the 4713 proteins of *S. Pullorum* R51, 564 proteins were evaluated for high antigenicity, and 230 proteins were chosen based on their high level of antigenicity in combination with the previous results (Appendix A).

### 3.6. Evaluation of Adhesive Probability

To interfere with the bacterial interaction with the host, targeting the adhesion probability of bacteria is crucial. Therefore, the adhesion probabilities of the protein candidates were analyzed using the SPAAN server, and only those with adhesion probabilities > 0.5 were considered. Based on these results, the 53 shortlisted proteins were selected for further analysis (Appendix A).

### 3.7. Prediction of Transmembrane Domains and Solubility

To facilitate in vivo studies, proteins with fewer transmembrane segments (<2) and high solubilities were prioritized for purification. First, TMHMM v.2.0 was used to predict the number of transmembrane segments and 52 proteins were shortlisted. Then, SoDoPE servers were used to filter the proteins based on solubility prediction, with 31 proteins showing a >0.5 probability of solubility being selected for further analysis (Appendix A).

### 3.8. Comparative Analysis

After performing a BLASTp analysis of the 31 candidates, it was found that 22 proteins showed significant sequence conservation with *S. Pullorum* virulence serovars (Appendix A) with substantial sequence coverage.

Based on surface localization, a high antigenic value, adhesin probability, transmembrane domains, and solubility, the 22 selected proteins can be considered as broad-spectrum vaccine candidates against *S. Pullorum*. The selected 22 candidate proteins and their functions are listed in Table 3.

### 3.9. Protein Expression and Purification

The 22 candidate genes were subcloned into pET-30a (Appendix A). Western blotting analysis confirmed that the 18 proteins with His-tags were expressed in the inclusion bodies of *E. coli* Rosetta cells after the induction with 1 mM IPTG at 16 °C for 12 h (Appendix A). The approximate molecular weights (Figure 3) after purification were consistent with their expected weights (Appendix A).

### 3.10. Immunogenicity Assay

To evaluate the immunogenicities of target vaccine proteins, animal experiments were conducted. A total of 114 eggs were randomly divided into 19 groups on the 18th day of incubation, and chick embryos were immunized with purified proteins. Blood samples were collected from chicks on day 14 post-immunization to evaluate the antibody response. The ELISA results demonstrate that PstS, SinH, LpfB, SthB, OmpC, PagN, and StiH elicited a significant antibody response compared to the control group (Figure 4). Inter- and intra-assay variabilities of the ELISA results are listed in Table 4.

### 3.11. Bacterial Loads after Challenge

The challenge test with the R51 strain in chicks was carried out to investigate the protective efficacy of the vaccine candidates. On day 11 post-hatching, all birds were orally administered 100 μL live R51 (1 × 10^7^ CFU/bird). The bacterial loads in the spleen and liver were measured on day 7 post-challenge. The results indicate that SinH, PstS, and LpfB have relatively fewer bacterial loads in the liver, and that SthB and LpfB have relatively fewer bacterial loads in the spleen than the control (Figure 5).

### 3.12. Evaluation of Protective Immunity

Based on previous experimental results, we selected four proteins (PstS, SinH, LpfB, and SthB) to evaluate protective immunity based on previous experimental results. Eighty eggs were randomly assigned to 5 groups (PBS, PstS, SinH, LpfB, and SthB) on day 18 of incubation. Embryos were immunized with purified proteins. The immunization process is the same as above. On day 11 post-hatching, all the birds were orally inoculated with 100 μL live R51 (1 × 10^9^ CFU/bird). The mortality of 16 birds in each group was monitored daily for 14 days.

The survival rate results indicate that PstS confers a significant protective effect against the R51 strain in poultry birds. The survival rate for SinH was 43.75%, 50% for LpfB, 62.5% for SthB, and a high rate of 75% for the PstS group, compared to 31.25% for the PBS control group (Figure 6). These findings suggest that PstS may be a promising candidate for further development as a protective agent.

## 4. Discussion

*S. Pullorum* is a pathogen that causes significant damage to the poultry industry in developing countries due to a high morbidity and mortality in young broilers. Additionally, adult broilers have a latent infection that is vertically transmitted to chicks through eggs [3]. Over the past century, enormous control measures have been taken to control such devastating diseases, including flock culling and antibiotic usage. However, the emergence of multidrug-resistant (MDR) *Salmonella* [78] necessitated using alternative strategies. Vaccination is the most effective method for preventing salmonellosis, but conventional vaccines are expensive with low immunogenicity [79]. Reverse vaccinology could be a more practical approach to identifying novel vaccine candidates, while such a method has not been conducted in mitigating Pullorum disease.

Surface-exposed proteins were identified as crucial vaccine targets due to their protective characteristics as virulence factors and the accessibility of antibodies to them [42,80]. However, available bioinformatics tools cannot predict protein localization, and some are classified as unknown [81]. Using multiple subcellular localization servers and signal peptide prediction significantly reduced the possibility of missing valuable vaccine candidates. In this study, out of 4713 proteins in *Salmonella* R51, 22 vaccine candidates were identified as potential targets. Subsequently, those candidates were validated through animal experiments, and the results indicated that representative proteins, including PstS, SinH, LpfB, OmpC, and PagN, could act as immunogens and stimulate chicks to produce specific antibodies.

The candidate vaccine antigenic proteins identified in this study participate in different biological processes of *Salmonella*. PstS, a phosphate ABC transporter, is a periplasmic phosphate-binding protein [75,76]. Phosphate is an essential nutrient for cell function and life. It is found in lipids, nucleic acids, proteins, and carbohydrates, and is involved in many biochemical reactions dependent on phosphoryl transfer [82]. The Pho regulon is a global regulatory circuit in bacterial phosphate management [83]. Previous studies have shown that the Pho regulon influences bacterial virulence, and in some bacteria, it directly controls virulence gene expression [84]. It has also been shown that the ABC transporter has good antigenicity and is an ideal vaccine candidate [75,85]. LpfA, LpfB, SthB, StiH, and StfF are all fimbriae-related proteins in the *Salmonella* genome. These are important virulence factors and promising vaccine antigens for *Salmonella* [86,87]. SinH, OmpC, and PagN are adhesive proteins crucial for bacterial invasion [58,60,88]. Additionally, SinH has also been identified as a promising vaccine target that reduces *E. coli* colonization and virulence. A previous study showed that the immunization of a murine host with SinH-based antigens elicited significant protection against various strains of the pandemic ExPEC sequence type 131 (ST131) and multiple sequence types in two distinct models of infection [89]. It is worth mentioning that OmpC and PagN have been confirmed as suitable immunogens that induce prominent immune responses in the host. A previous study showed that recombinant OmpC protein from *Salmonella* Typhimurium was immunized in a 4-week flock, and ELISA results showed that OpmC induced a significantly higher humoral immune response than the control. It is also primed a stable cell-mediated immune response at the same time. A protective index (based on fecal shedding of the organism) of rOmpC-based preparations ranging between 50 and 75% was observed for 3 weeks after the challenge [61]. According to a previous study, the detection of constant high titers of serum IgG and intestinal secretory IgA in immunized mice revealed that PagN (T2544) had been tested as a potential vaccine candidate in *Salmonella* Typhi. PagN antiserum increased macrophage uptake and the clearance of *Salmonella*, and enhanced complement-mediated lysis in vitro, indicating a role for T2544-specific antibodies in the killing process [33].

Altogether, our analysis shows that four antigens PstS, SinH, LpfB, and SthB can be appropriate candidates against Pullorum disease, which is worth further experimental investigation. Although only serum antibody rise cannot mediate complete protection, antibodies are essential in the clearance of the *S*. *Pullorum* infection and the induction of anti-*Salmonella*-protective Th1 responses. The ELISA test in this study revealed a significant rise in specific antibody titers in the sera of immunized poultry birds against PstS, SinH, LpfB, and SthB, as predicted by computational prediction, confirming that these vaccine candidates could be promising immunogens.

## 5. Conclusions

In this study, we employed a reverse vaccinology approach to identify potential vaccine candidates against *S. Pullorum*. According to epidemiological surveillance, computational pipeline, and animal experiments, we found that PstS, SinH, LpfB, and SthB could elicit a significant immune response and reduce the bacterial loads in immunized chicks. Particularly, PstS was confirmed as a potential vaccine candidate against *S. Pullorum*. Our findings illustrated that integrating in silico prediction and in vivo experiments could provide a compelling direction for vaccine development. The selected vaccine candidates, targeting critical virulence factors in *Salmonella* infection, make them promising candidates for further testing and potential use in the poultry industry.

## Figures and Tables

**Figure 1 vaccines-11-00865-f001:**
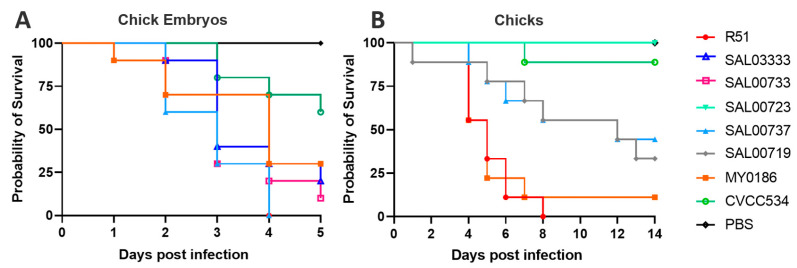
Virulence assays of *S. Pullorum* R51 and additional seven *S. Pullorum* clinical isolates in 16-day-old chick embryos and 1-day-old chicks. (**A**) Chick Embryos, (**B**) Chicks.

**Figure 2 vaccines-11-00865-f002:**
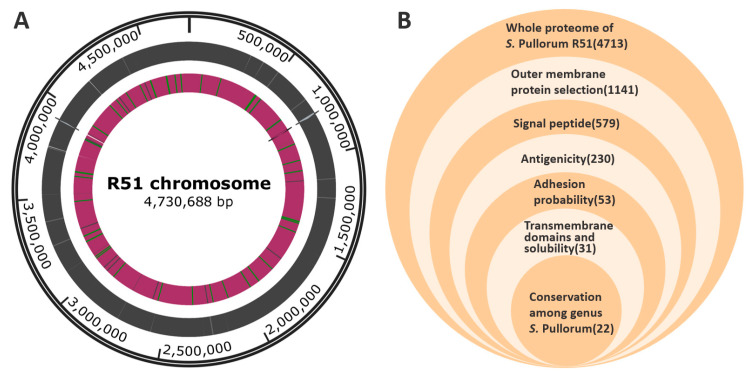
(**A**) Complete genome sequence of *S. Pullorum* R51; (**B**) Overview of the reverse vaccinology steps used in this study to predict potential vaccine candidates against the *S*. *Pullorum* R51 strain.

**Figure 3 vaccines-11-00865-f003:**
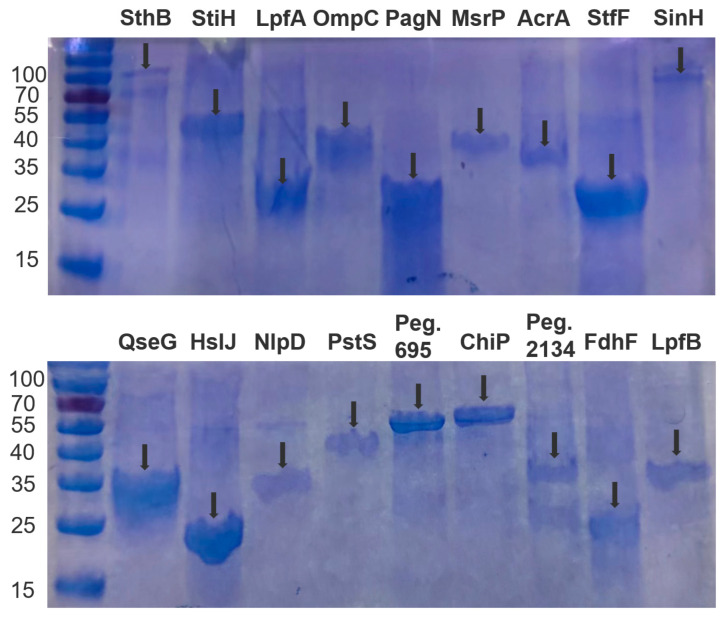
SDS-PAGE analysis of 18 recombinant target proteins expressed in *E. coli* Rosetta after purification. The arrows indicate the protein band for further studies.

**Figure 4 vaccines-11-00865-f004:**
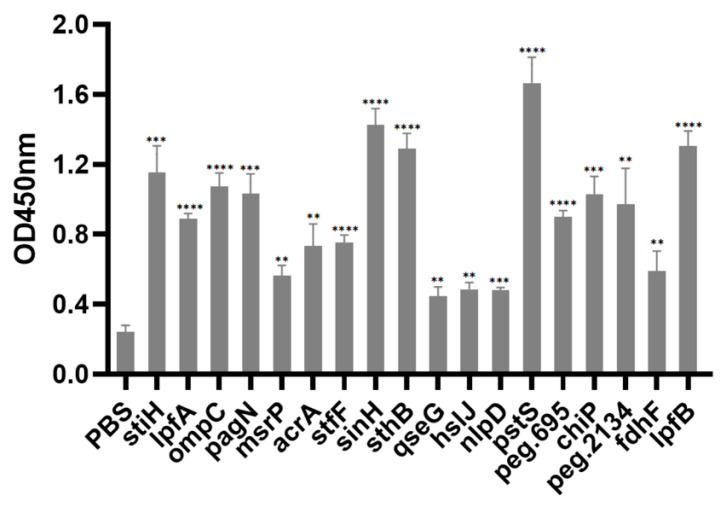
ELISA analysis of sera collected on day 14 post-immunization (day 11 post-hatching). The antisera from chicks mock-vaccinated with PBS were set as a negative control (NC). Statistical analysis was compared to the NC group. *p* < 0.01 (**), *p* < 0.001 (***), or *p* < 0.0001 (****).

**Figure 5 vaccines-11-00865-f005:**
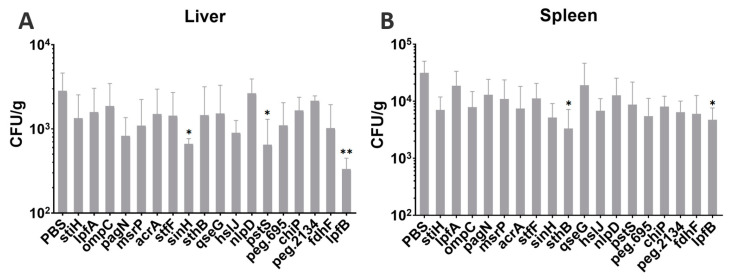
Bacterial loads in the spleen and liver were determined on day 7 after infection with *S. Pullorum* R51. Statistical analysis compared to NC group. *p* < 0.05(*), *p* < 0.01(**). (**A**) Liver, (**B**) Spleen.

**Figure 6 vaccines-11-00865-f006:**
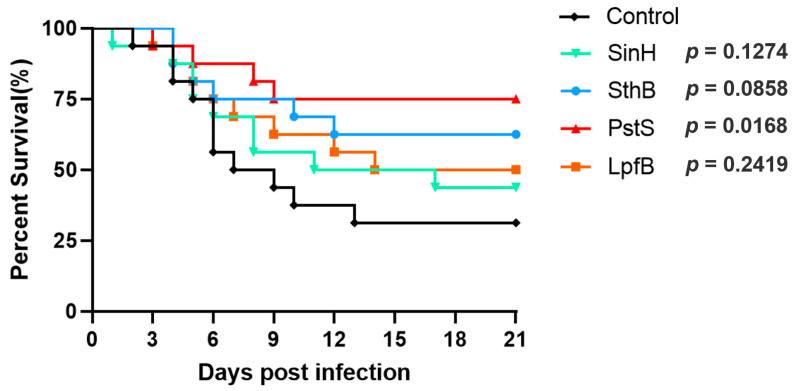
Survival percentage of PBS, SinH, SthB, PstS, and LpfB immunizing chicks infected with *S. Pullorum* R51. Survival of chicks was monitored for 21 days.

**Table 1 vaccines-11-00865-t001:** A list of programs and web servers used in the present study.

Function	Program	Web Address
Proteome acquisition	NCBI	https://www.ncbi.nlm.nih.gov/nuccore/nz_cp068386.1 (accessed on 10 February 2022)
Subcellular location	PSORTb	https://www.psort.org/psortb/ (accessed on 15 February 2022)
Subcellular location	CELLO2GO	https://cello.life.nctu.edu.tw/cello2go/ (accessed on 11 February 2022)
Subcellular location	BUSCA	https://busca.biocomp.unibo.it/ (accessed on 11 February 2022)
Subcellular location	SOSUI	https://harrier.nagahama-i-bio.ac.jp/sosui/ (accessed on 10 February 2022)
Signal peptide	SignalP-5.0	https://services.healthtech.dtu.dk/service.php?SignalP-5.0 (accessed on 16 February 2022)
Antigenicity	VaxiJen-3.0	https://www.ddg-pharmfac.net/vaxijen3/ (accessed on 18 February 2022)
Adhesin probability	SPAAN	https://github.com/3itamura-felipe/adhesin_finder (accessed on 22 February 2022)
Transmembrane domains	TMHMM-2.0	https://services.healthtech.dtu.dk/service.php?TMHMM-2.0 (accessed on 23 February 2022)
Solubility	SoDoPE	https://tisigner.com/sodope (accessed on 23 February 2022)
Conservations in Pullorum	BLASTp	https://blast.ncbi.nlm.nih.gov/Blast.cgi (accessed on 28 February 2022)

**Table 2 vaccines-11-00865-t002:** Quantity and proportion of predicted subcellular localization of *S. Pullorum* R51 proteins according to SOSUI, BUSCA, CELLO2GO, and PSORTb servers.

	SOSUI	BUSCA	CELLO2GO	PSORTb
Extracellular	157	438	144	72
	3%	9%	3%	2%
Outer Membrane	184	79	171	110
	4%	2%	4%	2%
Periplasm	361	NA	595	162
	8%	NA	12%	3%
Cytoplasmic	2565	3052	2875	2002
	54%	65%	61%	42%
Inner membrane	969	1107	928	1158
	21%	23%	20%	25%
Unknown	477	37	NA	1209
	10%	1%	NA	26%

**Table 3 vaccines-11-00865-t003:** Functional annotation of the potential vaccine candidates identified in *S. Pullorum* R51.

Candidates	Annotation
*dcrB*	DcrB is up-regulated by conditions that promote the production of known virulence factors [55]
*qseG*	QseG is an OM protein necessary for translocation of TTSS effectors [56,57]
*ompC*	OmpC is the main protein responsible for the antibody-mediated memory bactericidal response induced by porins [58,59]
*pagN*	Haemagglutinin that facilitates the adhesion to and invasion of epithelial mammalian cells [33,60]
*msrP*	MSR system is involved in the repair of periplasmic proteins oxidized by hypochlorous acid HOCl, which is generated in particular within phagocytic cells [61,62]
*sinH*	*sinH* encodes an autotransporter protein that facilitates adhesion and invasion into host cells [63,64]
*acrA*	AcrA protein is a component of multidrug efflux pumps, which can increase drug resistance [65,66]
*lpfA*	Long polar fimbria protein A precursor [67,68]
peg.1230	Putative fimbriae *stiH*
peg.1041	Fimbrial operons *sthB*
peg.3954	Minor fimbrial subunit StfF
*hslJ*	Heat shock protein, correlated to *E. coli* resistance to the antibiotic Nov [69]
*sodC*	Periplasmic superoxide dismutase plays a critical role in this survival by combating phagocytic superoxide [70,71]
*yhcN*	YhcN is involved in the response to hydrogen peroxide stress [72]
*nlpD*	A virulence factor that permits *S. Typhimurium* to survive under acid stress conditions [73,74]
*pstS*	Phosphate ABC transporter, periplasmic phosphate-binding protein PstS [75,76]
peg.695	Involved in the transport of maltose and maltodextrins
*chiP*	Involved in the uptake of chitosugars [77]
peg.2134	N-acetylneuraminic acid outer membrane channel protein NanC
*Fdh-2*	Formate dehydrogenase
Peg.1210	Probable secreted protein
*lpfB*	Chaperone protein *lpfB* precursor [68]

**Table 4 vaccines-11-00865-t004:** Inter- and intra-assay variabilities of candidate proteins.

Candidates	Inter-Assay		Intra-Assay 1		Intra-Assay 2		Intra-Assay 3	
	OD_492_	CV%	OD_492_	CV%	OD_492_	CV%	OD_492_	CV%
	Mean ± SD		Mean ± SD		Mean ± SD		Mean ± SD	
PBS	0.237 ± 0.015	6.31	0.231 ± 0.012	5.19	0.258 ± 0.018	6.98	0.223 ± 0.008	3.59
StiH	1.120 ± 0.084	7.46	1.217 ± 0.035	2.88	1.129 ± 0.032	2.83	1.013 ± 0.025	2.47
LpfA	0.889 ± 0.025	2.76	0.923 ± 0.019	2.06	0.880 ± 0.039	4.43	0.865 ± 0.024	2.77
OmpC	1.074 ± 0.063	5.89	0.986 ± 0.022	2.23	1.102 ± 0.042	3.81	1.133 ± 0.036	3.18
PagN	1.033 ± 0.092	8.92	0.905 ± 0.036	3.98	1.076 ± 0.033	3.07	1.118 ± 0.043	3.85
MsrP	0.564 ± 0.049	8.62	0.564 ± 0.028	4.96	0.504 ± 0.028	5.56	0.623 ± 0.039	6.26
AcrA	0.732 ± 0.104	14.21	0.876 ± 0.015	1.71	0.635 ± 0.035	5.51	0.684 ± 0.033	4.82
StfF	0.753 ± 0.035	4.60	0.801 ± 0.016	2.00	0.722 ± 0.017	2.35	0.735 ± 0.037	5.03
SinH	1.426 ± 0.076	5.33	1.376 ± 0.032	2.33	1.368 ± 0.034	2.49	1.533 ± 0.047	3.07
SthB	1.289 ± 0.073	5.66	1.324 ± 0.044	3.32	1.187 ± 0.022	1.85	1.355 ± 0.035	2.58
QseG	0.445 ± 0.044	9.82	0.483 ± 0.032	6.63	0.469 ± 0.019	4.05	0.384 ± 0.024	6.25
HslJ	0.485 ± 0.031	6.40	0.455 ± 0.031	6.81	0.528 ± 0.017	3.22	0.473 ± 0.019	4.02
NlpD	0.480 ± 0.014	2.87	0.472 ± 0.028	5.93	0.468 ± 0.024	5.13	0.499 ± 0.026	5.21
PstS	1.665 ± 0.120	7.20	1.597 ± 0.033	2.07	1.565 ± 0.039	2.49	1.834 ± 0.053	2.89
Peg.695	0.899 ± 0.030	3.27	0.897 ± 0.042	4.68	0.936 ± 0.033	3.53	0.864 ± 0.023	2.66
ChiP	1.029 ± 0.083	8.04	0.968 ± 0.013	1.34	1.146 ± 0.026	2.27	0.973 ± 0.035	3.60
Peg.2134	0.973 ± 0.167	17.19	0.856 ± 0.037	4.32	1.210 ± 0.021	1.74	0.854 ± 0.034	3.98
FdhF	0.589 ± 0.094	15.90	0.721 ± 0.043	5.96	0.514 ± 0.017	3.31	0.532 ± 0.033	6.20
LpfB	1.304 ± 0.071	5.41	1.372 ± 0.026	1.90	1.334 ± 0.032	2.40	1.207 ± 0.042	3.48

SD: Standard deviation, CV: Coefficient of variation.

## Data Availability

The data presented in this study are available in the Appendix A.

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
