# Peer review of "Identification and Evaluation of Novel Antigen Candidates against Salmonella Pullorum Infection Using Reverse Vaccinology"

_vaccines, 2023, doi:10.3390/vaccines11040865_

Round 1
Reviewer 1 Report
I think that the title well reflects the main aim and findings of the work.
I suggest to avoid the use of personal form (i.e. our, we etc.) throughout the study.
The abstract adequately summarize results and significance of the study, and, the keywords represent the article adequately. However, Authors should add some information on statistical analysis applied.
Throughout the text Authors should correct the scientific name of S. pullorum. Please make it in italic.
The introduction section is well written and it falls within the topic of the study, and Authors cited appropriately bibliographic information
The section of Materials and Methods is clear for the reader and it meticulously describes the methods applied in the study. However, some information should be added.
In the subsection 2.13. Immunogenicity assay, Authors should indicate the ELISA kit used (Manufacture and Country) and whether the used kit was specific for investigated species, or whether it was previously validated for the studied species. Moreover, for ELISA analysis, Authors should indicate the intra- and inter-assay variability.
Regarding statistical analysis, Did Authors perform a normality test in order to test the normal distribution of data? Please clarify this aspect.
Results section as well as Discussion section is clear and well written, the findings obtained in the study were well discussed and justified with appropriate references.
In the conclusion section Authors well summarize the main results of the study and emphasize the significance.
Tables and figures are generally good representing well the results gathered in the study.
Author Response
I think that the title well reflects the main aim and findings of the work.
Thank you for the positive feedback on the title of our work. We are glad that you think it reflects the aim and findings of our study.
I suggest to avoid the use of personal form (i.e. our, we etc.) throughout the study.
We appreciate your suggestion to avoid the use of personal forms throughout the study. We will ensure to make the necessary changes to the manuscript.
The specific changes are in Lines 22/77/151/219.
The abstract adequately summarize results and significance of the study, and, the keywords represent the article adequately. However, Authors should add some information on statistical analysis applied.
Thank you for acknowledging that our abstract adequately summarizes the results and significance of the study, and that the keywords represent the article well. We agree that adding information on the statistical analysis applied would enhance the quality and transparency of our study. We will revise the abstract to include a brief description of the statistical analysis. (Line 27)
Throughout the text Authors should correct the scientific name of S. pullorum. Please make it in italic.
We apologize for any errors in the scientific name of S. pullorum and will ensure to check it throughout the text and make it in italic. (Lines 28/249/290)
The introduction section is well written and it falls within the topic of the study, and Authors cited appropriately bibliographic information
We are pleased to hear that you found the introduction section well-written and relevant to the study. We take pride in citing appropriate bibliographic information.
The section of Materials and Methods is clear for the reader and it meticulously describes the methods applied in the study. However, some information should be added.
Thank you for noting that the Materials and Methods section is clear and meticulously described. We appreciate your feedback and will ensure to that additional information is added to provide more clarity for the reader. (Lines168-185)
In the subsection 2.13. Immunogenicity assay, Authors should indicate the ELISA kit used (Manufacture and Country) and whether the used kit was specific for investigated species, or whether it was previously validated for the studied species. Moreover, for ELISA analysis, Authors should indicate the intra- and inter-assay variability.
Thank you for your comment. We apologize for the confusion. We did not use an ELISA kit in our experiments. Instead, we used purified candidate proteins as the coating antigens for the immunogenicity assay. We will add the details of the relevant materials and the intra- and inter-assay variability to the manuscript. (Lines 199/203/313-315)
Regarding statistical analysis, Did Authors perform a normality test in order to test the normal distribution of data? Please clarify this aspect.
Thank you for raising the question about the normality test in our statistical analysis. We will include the results of the normality test for ELISA results in the table below to clarify the normal distribution of data. Thank you for pointing out this aspect.
Shapiro-Wilk |
StiH |
LpfA |
OmpC |
PagN |
MsrP |
AcrA |
StfF |
SinH |
SthB |
P-Value |
0.609 |
0.778 |
0.665 |
0.115 |
0.394 |
0.247 |
0.855 |
0.438 |
0.678 |
QseG |
HslJ |
NlpD |
PstS |
Peg.695 |
ChiP |
Peg.2134 |
FdhF |
LpfB |
PBS |
0.630 |
0.145 |
0.595 |
0.501 |
0.747 |
0.595 |
0.324 |
0.512 |
0.719 |
0.245 |
Results section as well as Discussion section is clear and well written, the findings obtained in the study were well discussed and justified with appropriate references.
We are pleased to hear that you found our Results and Discussion sections clear and well-written, with appropriate references to justify the findings of our study.
In the conclusion section Authors well summarize the main results of the study and emphasize the significance.
Thank you for your positive feedback on the conclusion section of our manuscript. We are glad that we were able to summarize the main results and emphasize the significance of our work.
Tables and figures are generally good representing well the results gathered in the study.
We appreciate your feedback on our tables and figures and are pleased to hear that they represent the results of our study well.

Reviewer 2 Report
Manuscript ID: Vaccines-2332491
Title: The authors used the process of antigen discovery based on genome information to identify antigens that may be of value for vaccine development against Salmonella Pullorum a causative agent of Pullorum disease of poultry.
Merit:
The experimental design is straightforward, and the approach has been successfully used in the discovery of protective antigens for vaccine development against pathogenic microorganisms.
Comments:
1. Lines 77-78. The statement that “antigenicity and efficiency of the optimal antigen candidates were further investigated in chick models” is unclear. What is meant by efficiency?
2. Lines 85-86: The primers used are not specific for S. Pullorum. Please clarify how the isolates were confirmed to be S. Pullorum.
3. Lines 92-93: Please clarify if the overnight culture was in liquid or solid media and the diluent used. How was it standardized to the 104 CFU/ml and confirmed to be that number?
4. It is not necessary to describe bioinformatic software used in detail as these are available in the public domain. Indicating the name of the program and what it was used for is sufficient. As an example, “to predict the subcellular localization of all proteins in S. Pullorum R51’s proteome, PSORTb v3.02 (42), BUSCA (43), SOSUI-GramN (44), and CELLO2GO (45) were utilized” is sufficient. The entire section on the selection of essential proteins should be re-evaluated and modified like this.
5. The manuscript is too long, and this is in part due to the presentation of unnecessary information mentioned above.
6. Aspects of materials and methods section needs improvement. Some areas lack specificity and clarity which will make reproducing the work by others challenging. As an example, lines 92-93 did not clarify if the overnight culture was in liquid or solid media and the diluent used. It also did not state how the 104 CFU/ml was determined and confirmed? This type of issue is present in multiple areas.
Overall, the work is good and publishable if the concerns raised are addressed.
Author Response
Manuscript ID: Vaccines-2332491
Title: The authors used the process of antigen discovery based on genome information to identify antigens that may be of value for vaccine development against Salmonella Pullorum a causative agent of Pullorum disease of poultry.
Thank you for your detailed comments on our manuscript. We are glad to hear that the purpose and methodology of our study were clear to you.
Merit:
The experimental design is straightforward, and the approach has been successfully used in the discovery of protective antigens for vaccine development against pathogenic microorganisms.
We appreciate your positive feedback on the straightforward experimental design and the success of our approach in identifying protective antigens for vaccine development against pathogenic microorganisms.
Comments:
- Lines 77-78. The statement that “antigenicity and efficiency of the optimal antigen candidates were further investigated in chick models” is unclear. What is meant by efficiency?
We apologize for the unclear statement in lines 77-78. By "efficiency" we meant the protective efficacy to protect the chick model against S. Pullorum. We will replace the term "efficiency" with "protective efficacy" to improve clarity in the revised manuscript. (Line 79)
- Lines 85-86: The primers used are not specific for S. Pullorum. Please clarify how the isolates were confirmed to be S. Pullorum.
Thank you for bringing this to our attention. The primers used were intended for initial screening purposes only, and confirmation of S. Pullorum isolates was done through DNA sequencing. We will include the size of the PCR product bands to clearly demonstrate their corresponding results. We apologize for any confusion in the original statement. (Lines 87-88)
- Lines 92-93: Please clarify if the overnight culture was in liquid or solid media and the diluent used. How was it standardized to the 104 CFU/ml and confirmed to be that number?
We apologize for the lack of clarity. The overnight culture mentioned in lines 92-93 was in liquid media, and the diluent used was sterile phosphate-buffered saline (PBS). The standardization of the culture to 104 CFU/ml was performed by serial dilution and plating on solid agar followed by colony counting. We will revise the manuscript to provide more specific details on the method used for standardization and confirmation of the bacterial load. (Lines 95-97)
- It is not necessary to describe bioinformatic software used in detail as these are available in the public domain. Indicating the name of the program and what it was used for is sufficient. As an example, “to predict the subcellular localization of all proteins in S. Pullorum R51’s proteome, PSORTb v3.02 (42), BUSCA (43), SOSUI-GramN (44), and CELLO2GO (45) were utilized” is sufficient. The entire section on the selection of essential proteins should be re-evaluated and modified like this.
Thank you for the feedback. We agree that providing only the name of the bioinformatic software used and its purpose is sufficient, as these tools are widely available in the public domain. We will revise the manuscript to provide a more concise and streamlined description of the bioinformatic tools utilized in our study. (Lines 123-124/134-135/140-141/144-146)
- The manuscript is too long, and this is in part due to the presentation of unnecessary information mentioned above.
We appreciate the feedback. We will carefully review the manuscript and eliminate any unnecessary information to ensure a more concise presentation of the results and conclusions.
- Aspects of materials and methods section needs improvement. Some areas lack specificity and clarity which will make reproducing the work by others challenging. As an example, lines 92-93 did not clarify if the overnight culture was in liquid or solid media and the diluent used. It also did not state how the 104 CFU/ml was determined and confirmed? This type of issue is present in multiple areas.
We acknowledge the concern. We will carefully review the materials and methods section and provide additional details, where necessary, to ensure clarity and specificity. We will also ensure that all relevant information, such as the type of media used, diluents, and methods for bacterial load determination, are clearly stated to facilitate reproducibility of our work. (Lines 168-185)
Overall, the work is good and publishable if the concerns raised are addressed.
Thank you again for your valuable comments, and we will address these concerns in the revised manuscript.
We appreciate the valuable inputs and suggestions by both reviewers, and believe the improved the manuscript can now be conditionally accepted. Please let me know if you need more information.
Best regards,
Min Yue, B.V.M., Ph.D.
Professor of Microbiology
Vice Chairman, Department of Veterinary Medicine
Vice Dean, The Rural Development Academy
Zhejiang University
866 Yuhangtang Road,
Hangzhou, Zhejiang Province 310058, P. R. China
